# From Pathogenesis to Intervention: The Importance of the Microbiome in Oral Mucositis

**DOI:** 10.3390/ijms24098274

**Published:** 2023-05-05

**Authors:** Julia S. Bruno, Ghanyah H. Al-Qadami, Alexa M. G. A. Laheij, Paolo Bossi, Eduardo R. Fregnani, Hannah R. Wardill

**Affiliations:** 1Instituto de Ensino e Pesquisa, Hospital Sírio-Libanês, São Paulo 01308-060, Brazil; juliasb9@gmail.com (J.S.B.);; 2School of Biomedicine, Faculty of Health and Medical Sciences, University of Adelaide, Adelaide 5005, Australia; 3Department of Oral Medicine, Academic Centre for Dentistry (ACTA), University of Amsterdam and Vrije Universiteit Amsterdam, 1081 LA Amsterdam, The Netherlands; 4Department of Preventive Dentistry, Academic Centre for Dentistry (ACTA), University of Amsterdam and Vrije Universiteit Amsterdam, 1081 LA Amsterdam, The Netherlands; 5Department of Oral Maxillofacial Surgery, Amsterdam UMC, University of Amsterdam, 1081 LA Amsterdam, The Netherlands; 6Department of Medical and Surgical Specialties, Radiological Sciences and Public Health, University of Brescia, 25121 Brescia, Italy; 7The Supportive Oncology Research Group, Precision Cancer Medicine Theme, The South Australian Health and Medical Research Institute, Adelaide 5000, Australia

**Keywords:** microbiota, stomatitis, biomarkers, radiotherapy, chemotherapy

## Abstract

Oral mucositis (OM) is a common and impactful toxicity of standard cancer therapy, affecting up to 80% of patients. Its aetiology centres on the initial destruction of epithelial cells and the increase in inflammatory signals. These changes in the oral mucosa create a hostile environment for resident microbes, with oral infections co-occurring with OM, especially at sites of ulceration. Increasing evidence suggests that oral microbiome changes occur beyond opportunistic infection, with a growing appreciation for the potential role of the microbiome in OM development and severity. This review collects the latest articles indexed in the PubMed electronic database which analyse the bacterial shift through 16S rRNA gene sequencing methodology in cancer patients under treatment with oral mucositis. The aims are to assess whether changes in the oral and gut microbiome causally contribute to oral mucositis or if they are simply a consequence of the mucosal injury. Further, we explore the emerging role of a patient’s microbial fingerprint in OM development and prediction. The maintenance of resident bacteria via microbial target therapy is under constant improvement and should be considered in the OM treatment.

## 1. Introduction

The ability of anti-cancer agents to effectively target and kill malignant cells relies mainly on their capacity to identify rapidly dividing cell populations. While this is a dominant attribute of malignant cells, many healthy tissues harbour subsets of highly proliferative cells, which are subject to the cytotoxicity of anti-cancer agents. The stem cells that populate mucosal surfaces of the body are highly vulnerable to collateral cytotoxicity, resulting in the formation of mucosal lesions [1]. The oral cavity is particularly susceptible to mucosal injury, leading to ulcerative lesions on the non-keratinised mucosa, such as the tongue, buccal mucosa, and lips, clinically referred to as oral mucositis (OM) [2].

OM is one of the most common, dose-limiting side effects in people undergoing systemic cytotoxic therapy and/or radiation for various cancers. However, it is particularly prevalent in stem cell transplant recipients treated with high-dose chemotherapy and people with head and neck cancer (HNC) receiving radiotherapy [2]. The profound breaches in the oral mucosa that occur in OM cause significant pain and impaired function (e.g., dysphagia, dysgeusia, difficulty in chewing), which collectively impact a patient’s ability to eat, drink and speak. As a result, severe OM is associated with a high reliance on enteral (tube) feeding (70–80%), analgesic use (80%) and even discontinuation of treatment (35%) [3,4]. As a result of its intensive supportive care needs, OM is associated with a considerable financial burden estimated at USD 5565 per chemotherapy cycle or USD 42,749 per patient after haematopoietic stem cell transplantation (HSCT) [3]. Given its profound impact on the patient’s quality of life and treatment adherence, OM has been the subject of intense research to understand its aetiology better and to develop preventive strategies. It is now well understood that while OM is initiated by direct DNA damage to rapidly dividing stem cells of the oral mucosa, it is perpetuated by aberrant and uncontrolled inflammation driven by dysregulated immune responses [5]. The intensity and persistence of these immune responses, which differ between patients based on their unique genetics, lifestyle factors and comorbidities, ultimately dictate the clinical severity, duration, and impact of OM [4,6].

The evolution in our understanding of OM’s pathogenesis has continued to grow and increase in complexity. Following the identification of aberrant immune responses in OM development, attention has turned to the oral microbiome—the collection of bacteria, viruses and fungi that inhabit surfaces and niches within the oral cavity [7]. Whilst these oral microorganisms have certainly been recognised in OM for decades, this has focused mainly on their opportunistic nature, with bacterial overgrowth and colonisation causing infections in ulcerated sites of the oral cavity [8]. However, more recently, the ability of the oral microbiome to causally affect and influence OM pathogenesis has been appreciated, and as a result, strategies to exploit it for clinical benefit have become an area of intense interest. In addition to the oral microbiome, emerging evidence suggests that the microbial community of the lower gastrointestinal tract, the gut microbiome, could also influence the course of OM development due to its critical role in modulating systemic immune responses. This review aims to provide an update on the current perspectives and evidence for how the microbiome causally contributes to OM, new evidence on its use as a biomarker and strategies to potentially modify the oral microbiome to influence OM.

## 2. The Oral Microbiome and Its Influence on Oral Health

The oral cavity is the second most colonised part of the human body, after the gut, yet it has a more diverse and dynamic microbial community [9]. While approximately 700 microbial taxa are described, a single individual harbours between 100 and 300 different microbial species [10]. The oral cavity consists of several distinct niches, including the teeth, gingival sulcus, tongue, cheeks, hard and soft palates, and tonsils [11]. Regarding microbial composition, *Streptococcus* is the most abundant genus in mucosal tissue, representing 44–55% of the detected genera, which can also be found in other specimens such as plaque and saliva. The genera *Neisseria*, *Prevotella* and *Haemophilus* are also highly abundant, with a prevalence of 6–29%, depending on the site. *Rothia* is also relatively abundant, with a prevalence of 4–24% in different sites, yet it is absent on the keratinised gingiva. Anaerobes such as *Actinomyces, Veillonella* and *Fusobacterium* are less prevalent and mostly present at subgingival sites [12].

Under healthy circumstances, the relationship between the microbiome and the host is in balance or symbiotic. A substantial change in the circumstances in the oral cavity can lead to a disbalance in the host-microbe interaction or dysbiosis. Oral microbial dysbiosis is associated with the development of several local oral conditions, including caries, periodontitis, and yeast infections, as well as an increase in the risk of systemic diseases such as diabetes, cardiovascular disease, and cancer [13]. It is currently known that single species do not cause oral disease but are instead associated with a shift in the oral microbiome from a predominance of health-associated species to a higher abundance of disease-associated components of the oral microbiome [14]. Both host and microbiome-derived factors maintain a healthy oral microbiome. Saliva and gingival crevicular fluid provide antimicrobial components (secretory IgA, lysozyme and lactoperoxidase) and nutrients for bacterial growth (e.g., carbohydrates and proteins). Oral bacteria exert both pro- and anti-inflammatory activity in interaction with the oral mucosa. However, how the balance between the oral microbiome and the host is maintained has yet to be fully understood [10].

The healthy balance between the host and the oral microbiome can be pushed into dysbiosis by several factors. Poor oral hygiene (biofilm accumulation on hard dental surfaces), diet, smoking, use of antibiotics and antimicrobial agents, low salivary flow rate, low activity of salivary proteins, genetic differences, malfunctioning of parts of the immune system and systemic diseases such as diabetes can act as central ecological pressure leading to dysbiosis of the oral microbiome [10]. The bacterial shift induced by cancer treatment may lead to changes in the homeostasis of the oral environment and, in turn, could exacerbate the severity of OM. First, the reduction in the abundance of commensal bacteria may facilitate the enrichment of pathogenic microorganisms. Commensal microbes can prevent the adhesion of transient and potentially pathogenic bacteria [15], induce epithelial cells to produce antimicrobial peptides [16,17], and favour organised cellular growth of the epithelial tissue [18]. In addition to the interaction with the mucosal epithelium, the metabolic potential of commensal bacteria is associated with a wide range of host benefits [19] and control of oral diseases [20] due to their ability to regulate pH and decrease oxidative stress [21].

In contrast, disease-associated bacteria present prominent virulence factors, such as lipopolysaccharide (LPS), fimbriae and proteolytic metabolites. The fimbriae expression is correlated with polymicrobial adhesion, colonisation, and invasion of host cells. This Gram-negative structure induces the expression of interleukin (IL)-1β, IL-6 and tumour necrosis factor (TNF)-α [22], which are also strictly linked to the development of oral mucosal inflammation [23]. Gram-negative bacteria also form outer membrane vesicles (OMV) to exteriorise metabolites. In addition, these structures favour the aggregation of other pathogens such as *Staphylococcus, Actinomyces, Treponema,* and *Tannerella* [24]. The periodontal pathogen *Porphyromonas* produces gingipains (highly active proteases) capable of destabilising epithelial cells and increasing epithelial permeability [23]. In addition to tissue integrity, gingipains are also associated with pro-inflammatory mechanisms, such as the induction of expression of inflammation-related factor cyclooxygenase (COX)-2 and production of prostaglandin E2 (PGE2) through the activation of ERK1/2 kinases and IκB in monocytes [25]. Interestingly, the co-interaction between *Fusobacterium* and *Porphyromonas* has been found to induce a greater activation of inflammatory pathways and potentiates the tissue-invasive capacity of *Porphyromonas* [26], suggesting a synergistic interaction between pathogens to colonise the environment, and as a side effect, it may aggravate/cause disease in the host.

## 3. The Oral Microbiome and Oral Mucositis: Cause or Consequence?

### 3.1. Dynamic Changes in the Oral Microbiome in the Setting of Oral Mucositis

Exposure to cytotoxic cancer treatments is associated with significant alterations in the oral microbiome; however, dissecting cause and consequence has proven to be exceptionally difficult. This complexity reflects the many co-occurring insults to the microbiome and the intimate bi-directional communication between the host and microbes. Potential causes for microbial changes include the direct antimicrobial effects of cancer therapies or the pathological changes in the oral cavity, such as the reduction in saliva production, disruption of the mucosal barrier, immune activation, and inflammation [27]. Several studies have characterised the shifts in the oral microbiome following exposure to different cancer treatments, including radiotherapy, chemotherapy, and stem cell transplantation (SCT) (Table 1). For example, Laheij et al. carried out a longitudinal analysis of the oral microbiome of 50 allogeneic SCT recipients pre-SCT and up to 18 months post-SCT. Their analysis revealed that alpha diversity was significantly reduced one week post-SCT and only recovered to the pre-SCT diversity level three months post-SCT. Lower alpha diversity was also associated with a lower abundance of the health-associated genera *Streptococcus* and *Veillonella* and an increased abundance of the disease-associated genera *Staphylococcus* and *Mycobacterium* post-SCT [28].

Similarly, a reduction in microbial diversity and compositional changes has been observed following radiotherapy, with changes observed in several oral taxa, including *Lactobacillus, Streptococcus, Gemella, Leptotrichia, Neisseria, Capnocytophaga, Neisseria, Olsenella, Parviomonas, Tannerella* and *Capnocytophaga* [29,30,31] (Table 1). Importantly, however, there is significant variability in the specific microbial changes reported across these studies, even across comparable patient populations. The resulting variations likely reflect the unique microbial communities that inhabit different geographical regions of the mouth, as well as variations in sampling and analytic methods and treatment time points.

Given the variable nature of the compositional analysis, mainly when performed and compared at defined or limited time points, the resilience of the oral microbiome and the speed of recovery could offer more reproducible indicators for the impact of the oral microbiome on OM. Recent studies have reported that better OM outcomes are associated with a more resilient oral microbiome, that is, a microbiome that is more stable over time following exposure to cytotoxic therapy [32,33].

**Table 1 ijms-24-08274-t001:** Literature review of microbial dynamics in patients with oral mucositis using next-generation sequencing methods.

Author	Study Population (*n*), Treatment	Sample Collection Site	Time Point (s)	Key Findings
Zhu et al., 2017 [34]	NPC (*n* = 41), CRT	Retropharyngeal	During RT (10, 20, 30, 40, 50, 60 and 70 Gy)	Developed ≥ grade 2 OM During severe OM: ↑ *Actinobacillus, Mannheimia and Streptobacillu*s During mild OM: ↑ *Enhydrobacter, Schwartzia, Pseudoramibacter, Treponema*
Vesty et al., 2020 [31]	HNC (*n* = 19), RT	Unstimulated whole saliva Oral mucosa	Pre-RT and during RT (0–20 Gy, 21–40 Gy and 41–60 Gy)	Developed ≥ grade 2 OM Baseline: ↑ *Capnocytophaga, Neisseria, Olsenella, Parviomonas, Tannerella*
Hou et al., 2018 [30]	HNC (*n* = 19), CRT	Oropharyngeal mucosa	Pre-RT and during RT (10, 20, 30, 40, 50, 60, and 70 Gy)	Developed ≥ grade 2 OM OM onset: ↑ *Prevotella, Fusobacterium, Treponema, Porphyromonas*
Reyes-Gibby et al., 2020 [35]	HNC (*n* = 57), CRT	Buccal mucosa	Pre-RT, OM onset and OM ulceration	Onset of OM non-ulcerated ↑ *Prevotella, Fusobacterium, Streptococcus* Development of severe OM Baseline: ↑ *Cardiobacterium, Granulicatella* During OM: ↑ *Megasphaera, Cardiobacterium*
Al-Qadami et al., 2023 [36]	HNC (*n* = 20), CRT	Stool	Pre-treatment	Developed ≥ grade 3 OM Baseline: ↑ *Eubacterium, Victivallis, Ruminococcus* Developed grade 0–1 OM Baseline: ↑ *Alistipes*
Hong B-Y et al., 2019 [27]	Solid tumours * (*n* = 49), 5-FU or doxorubicin QT	Unstimulated saliva Oral mucosa	Pre-QT, 3 days, 9 days, 14 days after QT infusion	Developed ≥ grade 2 OM During OM: *↑ Fusobacterium, uncultured* *Clostridiales, Prevotella, Treponema*
Laheij et al., 2019 [33]	MM (*n* = 51), auto-HSCT	Oral cavity rinse	Pre auto-HSCT, 0–4 days, 1 w, 4 w and 3 m after SC infusion	Developed ≥ grade 2 OM Baseline: ↑ *Veillonella, Enterococcus, Streptococcus, Staphylococcus, Fusobacterium, Prevotella* No OM development Baseline: ↑ *Streptococcus, Actinomyces*
Shouval et al., 2020 [37]	Haematological diseases (*n* = 604), allo-HSCT	Saliva	Pre allo-HSCT and 1w, 2 w, 3 w, 4 w, 5 w after SC infusion	Developed ≥ grade 0–1 Baseline: ↑ *Aggregatibacter, TG5, Lactobacillus, Butyrivibrio, Treponema, Schwartzia, Paludibacter* Developed ≥ grade 2–3 Baseline: ↑ *Kingella, Atopobium, Haemophilus, Fusobacterium, Corynebacterium, Actinomyces, Cardiobacterium* Developed ≥ grade 0–1 During OM: ↑ Filifactor, *Selenomonas, Brachymonas, Eikenella, Treponema*, *TG5* Developed ≥ grade 2–3 During OM: ↑ *Methylobacterium*
Bruno et al., 2022 [38]	Haematological diseases (*n* = 30), allo-HSCT	Buccal mucosa	Pre allo-HSCT, OM ulceration onset and OM healed	Developed ≥ grade 2 OM OM onset: ↑ *Mycoplasma* and *Lactobacillus* Post OM: ↑ *Staphylococcus, Treponema 2, Enterococcus, Lactobacillus*
Laheij et al., 2022 [28]	Haematological diseases (*n* = 50), allo-HSCT	Oral cavity rinse	Pre-SCT and 3 m, 6 m, 12 m and 18 m after SC infusion	Developed ≥ grade 2 OM OM onset: ↑ *Mycobacterium, Staphylococcus, Enterococcus*

NPC: Nasopharyngeal Carcinoma; CRT: Chemoradiotherapy; RT: radiotherapy; * Squamous-cell carcinoma, Breast cancer and Adenocarcinoma; MM: Multiple Myeloma; Auto-HSCT: Autologous Haematopoietic Stem Cell Transplantation; SC: Stem Cell; Allo-HSCT: Allogenic Haematopoietic Stem Cell Transplantation; w: week(s); m: month(s); ↑: enrichment or increased abundance.

### 3.2. Evidence of Causal Involvement of the Oral Microbiome in Oral Mucositis

The changes in the oral microbiome are relatively well described in the literature, albeit with variation in the specific microbial changes reflecting the heterogeneity of patient populations studied and methodologies employed. Although it remains unclear, it is speculated that these changes occur not as a direct result of chemotherapy or radiotherapy but instead in response to mucosal injury, which creates a hostile micro-environment for resident microbes [27]. This hostility is not only physical due to the destruction of mucosal niches within which bacteria reside but also oxidative and inflammatory, and capable of damaging anaerobic and sensitive commensal microbes. Of note, many pathogenic microbial taxa are more robust, having developed mechanisms to survive in such hostile environments, hence their subsequent expansion and domination of the oral cavity in the context of OM.

The complex and bi-directional communication between the host and their resident microbes, as well as the numerous factors that impact microbial stability in cancer therapy (e.g., antibiotics, dietary changes, stress, medications), has made it difficult to determine if the dysbiotic oral microbiome actually contributes to OM pathogenesis, or is simply an innocent bystander that changes in response to OM. In determining the causal contribution of the oral microbiome in OM development, the use of germ-free mice is the most robust approach. Germ-free mice are devoid of a microbiome, and thus their response to drugs or challenges when compared with conventional mice (i.e., those with a microbiome) can be used to determine the causal role of the microbiome in certain diseases or functions. In a recent study using germ-free mice, OM induced by 5-FU chemotherapy was significantly less severe than in specific pathogen-free (SPF) conventional mice. Notably, this decreased sensitivity was accompanied by marked blunting of matrix metalloproteinase (MMP)-3 and MMP-9 production and lower cytokine gene expression [39]. To date, this is the only study to use germ-free mice in understanding OM and, as such, is a landmark study. While their results underscore the potential causal role of the oral microbiome in OM, they must be interpreted cautiously. Firstly, this does not confirm that the oral microbiome is augmenting OM, as the gut microbiome is depleted in germ-free mice. The oral–gut microbiome axis has been hypothesised to contribute to OM by its ability to augment systemic immune responses and drug metabolism [40].

Similarly, this study does not provide substantial mechanistic insights as the mice were terminated at just a single time point; as such, it is unclear if the oral microbiome dictates the development of OM or its capacity to recover. Most importantly, while germ-free models are the gold standard method for understanding causal microbial mechanisms, germ-free mice are inherently different to conventional mice, with numerous immune impairments that result from no exposure to microorganisms throughout life [41]. To eliminate this confounding factor, colonising germ-free mice with microbes to “conventionalise” them is ideal.

An alternative approach to determining the causal role of the microbiome on OM pathogenesis is mimicking the germ-free setting through antibiotic-induced depletion. This approach was adopted by Al-Qadami et al., 2022 [42] in a rat model of radiation-induced OM. The microbiome was depleted using a cocktail of antibiotics (ampicillin, neomycin, and vancomycin) administered in drinking water, and depletion was confirmed through 16S rRNA gene sequencing. Critically, this study focused on the depletion of the gut microbiome, but given the administration of antibiotics in drinking water, it is plausible that the oral microbiome was depleted given the systemic uptake of these antimicrobials. Antibiotic-depleted rats that received irradiation to the snout developed less severe OM, with fewer days of severe OM owing to faster mucosal recovery in the healing phase. This aligns with similar findings for chemotherapy-induced gastrointestinal mucositis, with the gut microbiome causally contributing to mucosal recovery (not its initial susceptibility to injury) [43]. While it cannot be concluded whether this was explicitly the oral or gut microbiome, especially in light of data that show antibiotics administered in drinking water have limited effect on the oral microbiome [44], these findings support the hypothesis that the alimentary (i.e., oral and gut) microbiome causally contributes to OM. Of interest, these data differ from the findings of a double-blind, randomised controlled trial (RCT) investigating the effect of selective elimination of oral bacteria on OM in people receiving head and neck radiotherapy [45]. Specific antimicrobial agents delivered via lozenges (for the entire course of radiotherapy) did not affect OM scores in the first five weeks of treatment. This either suggests that: (i) the effects seen preclinically (i.e., in germ-free mice and antibiotic-depleted mice) are dictated by the gut microbiome, (ii) it is the loss in oral commensal microbes (not the expansion of pathogens) that contributes to OM, (iii) the antimicrobial agents targeted the ‘wrong’ microbes, both commensal and pathogenic or (iv) the effects of the oral microbiome are restricted to the healing phase of OM pathogenesis. Without data from beyond five weeks of treatment, it is not possible to draw robust conclusions. However, given recent preclinical data indicating the beneficial effects of probiotic supplementation for OM caused by 5-FU, the role of the oral–gut microbiome axis cannot be ignored [46]. Similar results have been seen clinically, with a probiotic cocktail shown to augment adverse systemic immune responses and reduce OM induced by concurrent radiotherapy and chemotherapy for nasopharyngeal carcinoma [47]. Similarly, distinct gut microbiome signatures have been shown to differ between HNC patients with and without OM [36].

It is, therefore, likely that the oral and gut microbiome work in concert to regulate OM pathogenesis, a concept supported by a recent study in which oral microbiome transplantation restructured oral and gut bacteria configurations and programmed gene expression profiles of the tongue to mitigate OM caused by radiation [48].

### 3.3. Mechanisms by Which the Microbiome Contributes to Oral Mucositis

Although it is yet to be confirmed, there is a range of mechanisms by which the oral- and gut-microbiome may causally contribute to OM development. These largely centre on their ability to influence many aspects of mucositis aetiology, including drug metabolism, innate immune responses, oxidative stress and cellular recovery and repair, each of which has been extensively reviewed and discussed (for an in-depth discussion of these mechanisms, please see [40,49,50,51]). Ultimately, via its regulation of drug metabolism, the microbiome can directly influence the circulating concentrations of chemotherapeutic drugs and, therefore, their capacity to induce mucosal injury; this has been particularly well demonstrated in the context of irinotecan treatment [52]. In parallel, by influencing (both positively and negatively) the host’s immune responses, the microbiome (both oral and gut) can modulate the depth of mucosal injury and its capacity to recover. For example, when the pattern recognition receptor TLR4 (which enables pathogenic microbes to trigger damaging immune responses) is deleted, mucositis and its associated symptoms are minimised [53].

When considering these mechanisms, it is important to acknowledge the nuances of the oral cavity, both in terms of its epithelial variations and architectural differences. For example, variations in keratin thickness, cellular proliferation, cell adhesion and the number of epithelial layers dictate tissue resistance/sensitivity to injury. These, combined with the specifics of cytotoxic therapy, will undoubtedly influence OM development. For example, chemotherapy-induced OM is more prevalent in regions of the lining (non-keratinised) mucosa (e.g., buccal mucosa, labial mucosa, ventral tongue surface and soft palate), while patients receiving radiotherapy to the head and neck region tend to experience OM that reflects the irradiation field, regardless of keratin presence [54]. Therefore, we may presume that areas of greater structural resistance (keratinised mucosa) require more impactful damage for the development of an ulcer, and thicker epithelium (lining, non-keratinised mucosa) is more susceptible to drug toxicities [55].

In addition to the physical attributes of the oral cavity influencing the susceptibility to direct cytotoxic injury, these regions and structural attributes also influence how the oral cavity, and by extension, the host immune system interacts with the microbiome [56]. The innate immune response is the ancient pathogen recognition response in the human body and part of the early progression of OM [57]. Immune response receptors, which mediate the host’s response to invading pathogens, are located throughout the alimentary tract, including the oral cavity [56]. These include leucine-rich repeat-containing receptors (NLR), protease-activated-receptor (PAR) and, probably the best-described family of receptors, toll-like receptors (TLRs) [23,56] (Figure 1). Among the ten receptors in the family, the most microbiome-correlated are TLR2, TLR4, TLR5, and TL9 [54,58].

Commensal bacteria from the oral cavity interact with immune receptors to maintain immunological homeostasis [16,17]. Nuclear factor kappa B (NF-ĸB), the “gatekeeper” of OM, is activated by the binding of pathogen-associated molecular patterns (PAMPs) to TLRs [23,56] (Table 2). TLR2 recognises bacteria in the basal cell layer and acts as a physical barrier when connected to glycoproteins, preventing the entry of harmful substances into the cell [23,56]. TLR4 initiates pro-inflammatory cascades and up-regulates the antigen-presenting of immune cells in response to binding of the lipopolysaccharide—a cell wall product of Gram-negative bacteria (LPS) [56]. This represents a key causal mechanism by which the oral microbiome can exacerbate OM and extend the duration or severity of the injury.

Of interest, though, is the heterogeneity in LPS, with different LPS subtypes capable of modulating different receptors [59]. Heterodimer TLRs such as TLR1 and TLR6 with TLR2 function as receptors for atypical LPS, such as those formed by Leptospira and P. gingivalis which are structurally different from Gram-negative LPS. When there is excessive contact by LPS with commensal bacteria, immune protective actions, such as receptor internalisation, occur [23]. In contrast to TLR4, other TLRs recognise different microbial compounds, such as flagellin, which is recognised by TLR5 [60]. An in vivo study demonstrated that a pharmacological agonist of TLR5, called CBLB502, influenced OM severity and accelerated regeneration through the production of growth factor (G-CSF) and superoxide dismutase (SOD2) [56,61]. This highlights TLRs’ causal, yet contradictory, roles in controlling how the oral microbiome impacts OM.

## 4. Clinical Applications of the Microbiome in OM

The oral and gut microbiome are undeniably involved, to some extent, with the development and presentation of OM. However, as we have outlined, dissecting their causal effect has been extremely difficult. Despite this complexity, accumulating data support the idea that the microbiome can be used as a tool to improve OM outcomes in patients with cancer, thus, supporting a clinically meaningful causal role [37]. In addition, it is becoming increasingly clear that the highly individualised nature of both the oral and gut microbiomes (which has been likened to a fingerprint) could be used as a predictive or diagnostic biomarker.

### 4.1. Microbial Fingerprints as Biomarkers for Oral Mucositis

The human microbiome, including the oral and gut microbiome, is shaped by various factors such as genetics, lifestyle, and diet, and each person harbours a unique microbial community that reflects the highly unique combination of endogenous and exogenous factors. The temporal dynamics of the oral and gut microbiome differ more significantly between individuals than within individuals, and each individual has a distinct microbial fingerprint [62].

An individual’s unique microbial fingerprint is increasingly recognised for its capacity to influence their risk of disease and response to various medications, including cancer treatments. The oral microbiome has been shown to predict the risk not only of oral diseases (e.g., dental caries, periodontitis and oral cancer) but also distant and systemic conditions such as gastric, pancreatic and colorectal cancers [63,64,65]. A machine-learning model based on the oral microbiome composition was able to predict the risk of developing dental caries in early childhood with an area under the curve (AUC), an indicator of model accuracy, of 0.71 (2 years before caries onset) and 0.89 (immediately before caries diagnosis) [66]. Similarly, another study used an oral microbiome panel to develop a model to identify individuals with oral and oropharyngeal cancer (AUC: 0.98) [67]. Systemically, studies have established oral microbiome-based prediction models to distinguish gastric cancers from non-malignant gastric lesions (AUC: 0.91) [63] and colorectal cancers from healthy controls (AUC: 0.90) [65].

Similarly, oral microbiome-based models can be used to predict the risk and severity of OM using the baseline/pre-therapy or post-therapy oral microbiome. Despite the theoretical potential of this approach, only a few studies have been conducted to assess the feasibility of using such models in the context of OM. Zhu et al. analysed the oral (mucosal) microbiome of 41 patients with nasopharyngeal carcinoma treated with radiotherapy or chemoradiotherapy. The samples were collected before irradiation and between day 5 and day 35 (at five-day intervals) during irradiation. The results demonstrated that, following treatment initiation, as OM severity increases, the oral microbiome community of patients who developed severe OM (grade ≥ 2) became more distinguished from those who developed less severe OM (grade 0–1) and healthy controls prior to therapy. Interestingly, analysing the microbiome from OM lesions early in their development (Grade I/II) showed that those who progressed to develop more severe OM had lower alpha diversity and a high abundance of Gram-negative rods (*Streptobacillus*, *Actinobacillus* and *Mannheimia*). The authors then used a random forest model to compare the oral microbiome using baseline (non-irradiated) samples, Grade 0 OM samples and Grade I/II OM samples. While the use of baseline and Grade 0 OM samples yielded low accuracy models (AUC: 0.64 and 0.65 respectively), using samples collected at grade 1–2 OM produced a more accurate predictive model (AUC: 0.89) that can differentiate between those who progressed to severe OM and those who had stable mild OM throughout treatment. This suggests that oral microbiome at the early stages of OM can predict the clinical course of OM and determine whether the patient will develop more severe OM at later stages of the treatment [34].

Another study by Bruno et al. assessed the oral microbiome of mucosal swabs collected from 30 allogeneic SCT recipients before conditioning (preconditioning), at ulcerative OM onset and at ulcerative OM healing time points. The analysis revealed a dynamic change in microbial alpha and beta diversity throughout the course of OM, with the lowest alpha diversity observed at the OM healing time point. Furthermore, the relative abundance of *Porphyromonas* in the preconditioning samples was positively correlated with ulcerative OM, while, at ulcerative OM onset, a higher relative abundance of *Lactobacillus* was associated with a shorter duration of ulcerative OM. Additionally, the study established a support vector machine (SVM) model using eight genera from the preconditioning samples which could predict the OM onset with 96.6% accuracy [38].

In addition to OM onset and severity, one study has investigated the impact of the oral microbiome on OM healing months post-treatment completion. Jiang et al. analysed the oral mucosal microbiome of 64 patients with nasopharyngeal carcinoma to assess the association between the oral microbiome and OM healing six months post-treatment. Patients were divided based on the WHO OM scoring system into three groups: normal healing (Grade 0), mild delay in OM healing (Grade I/II) and severe delay in OM healing (Grade III/IV). The results demonstrated that the severe OM healing delay group had a higher abundance of Actinobacteria phylum and Veillonellaceae, Actinomycetaceae families and *Veillonella* genus. The study also found that two genera, *Actinomyces* and *Veillonella*, can be used as predictive markers (AUC of 0.96 and 0.82, respectively) for severe delay in OM healing [68].

Finally, Reyes-Gibby et al. utilised a mixture cure model to generate hazard ratios for OM development based on the oral microbiome community. In this study, buccal mucosa samples of 66 patients with head and neck squamous cell carcinoma were collected at baseline, immediately before any grade OM onset and immediately before severe OM onset. At baseline, a high abundance of *Cardiobacterium* and *Granulicatella* was associated with the risk of early onset of severe OM. Moreover, immediately prior to OM development, a higher abundance of *Streptococcus* was associated with delayed onset of severe OM, while a higher abundance of *Fusobacterium* and *Prevotella* was linked to early severe OM development. Additionally, immediately before severe OM onset, an increased abundance of *Cardiobacterium* and *Megasphaera* correlates to a higher risk of early severe OM onset [35]. These data indicate that oral microbiome-based models have the potential to be used as a biomarker for OM, with the potential to predict people at risk of developing OM. However, it is important to note that most studies have only been able to demonstrate meaningful results using microbial data during OM development, not before cancer therapy has started. Identifying strategies to increase the predictive power of the baseline microbial community is clinically necessary to identify high-risk OM patients. These patients may be offered tailored treatment programs or proactive, supportive care to minimise OM development and impact. It is likely that oral microbiome fingerprints will need to be used in conjunction with other conventional OM risk factors (e.g., genetics, comorbidities, medications) to create an integrated risk model.

Furthermore, given that the oral microbiome is influenced by multiple biological and environmental factors [69,70], several considerations need to be considered when using oral microbiomes as biomarkers. This includes sample collection (time point and method), sampling site and microbiome analysis techniques. The use of standardised protocols for oral microbiome sampling and analysis and the collection of the relevant clinical metadata are essential for establishing oral microbiome as a reproducible and reliable biomarker for OM. In addition to the oral microbiome, it is also relevant to consider the gut microbiome for its clinical use as a biomarker of OM. The gut microbiome has a larger and more stable microbial community than the oral microbiome and, therefore, can influence systemic inflammatory responses (and thus OM) with greater magnitude. Although a strong evidence base exists linking the composition of an individual’s gut microbiome with a variety of intestinal and systemic conditions [71,72], there are limited data in the context of OM. Currently, only one study has reported an association between the gut microbiome and the risk of developing severe OM. Al-Qadami et al. analysed the faecal microbiome of 20 patients (only 17 patients of these were included in OM analysis) with head and neck cancer treated with radiotherapy or chemoradiotherapy. They reported no significant difference in alpha and beta diversity between those who developed mild (Grade I/II) and severe OM (Grade III/IV). However, the higher relative abundance of three genera (*Victivallis*, *Eubacterium* and *Ruminococcus*) was associated with the development of Grade III/IV OM [36]. This study did not investigate the potential of using gut microbiome-based models to predict OM severity, likely due to the limited sample size. Given the profound heterogeneity in gut microbiome composition, the sample size needed to power these approaches is significant. It, therefore, highlights the need for international collaboration and leadership to deliver meaningful findings that hold relevance on a global scale.

### 4.2. Microbial Therapeutics for OM Prevention and Treatment

Given the microbial disturbances observed in the oral cavity during the course of OM, and the difference in composition at pre-treatment/baseline among patients who have severe/ulcerated OM and patients with milder symptoms, microbial-based therapies hold promise in minimising the burden of OM.

One of the most used yet nonspecific microbial therapies for OM is chlorhexidine (CHX), a biocidal agent with membrane destruction action on bacteria and fungi [73]. CHX is prescribed in patients at risk of OM to decrease the load and potentially contain the inflammation and degree of bacterial OM [74]. To date, there are no studies that prove the effectiveness of using CHX alone in the clinical course of OM. Its use is also undermined by CHX-related side effects [8]. Huang et al. retrospectively analysed 13,969 patients with head and neck cancer, of whom 482 patients were treated with 5-FU. It was noted that performing the periodontal procedure associated with the use of CHX increased the incidence of OM in patients treated with 5-FU [75]. Importantly, the MASCC/ISOO Clinical Practice Guidelines for OM prevention and management generally advise against the use of antimicrobials, including CHX, to prevent OM citing insufficient evidence of clinical benefits and possible collateral effects in HNC patients, such as intense taste alterations [76,77]. This highlights the fact that these strategies generally induce profound ecologic stress on the microbial environment, depleting commensal (beneficial) microbes as well as pathogens. Commensal microbes are critical in restricting pathogen growth whilst also promoting epithelial health, via the production of beneficial microbial metabolites (e.g., acetate, lactate) [73,78]. This was shown by Bescos et al. [79], who evaluated the effect of using 0.2% CHX for 1 min twice a day, for seven days, in 36 healthy patients and noted a decrease in salivary buffering capacity, lower nitrate-reducing capacity, lower alpha diversity and increases in *Neisseria*, *Streptococcus* and *Granulicatella*. Correlations between the higher abundance of specific genera with metabolites were reported: *Fusobacterium* and higher glucose concentration; *Actinobacteri*a and lower lactate concentration; and *Proteobacteria* and lower nitrite concentration.

In addition to their limited efficacy, antimicrobial strategies for OM are also unattractive as they contradict global antibiotic stewardship efforts, which aim to reduce the reliance on and use of antimicrobials and associated resistance. As such, efforts to support a “healthy” microbiome have shifted away from directly targeting pathogens to instead promoting commensal populations. Probiotics consist of live bacteria, possibly genetically mutated to produce metabolites, that provide benefits to the host [80]. The use of probiotics for personalised treatment aiming to reduce bacterial-dependent diseases, such as periodontal disease, is constantly improving [81,82], and the combination with other therapeutics, such as ozone and photobiomodulation, may be beneficial for decreasing the bacterial load [83].

Probiotics in people with cancer have generally been approached rather cautiously due to the perceived risk of infection in immunocompromised hosts. Despite this, a growing number of reports exist on their use for OM prevention and/or treatment. For example, in patients with nasopharyngeal carcinoma, Xia et al. [47] evaluated, through a phase II RCT, the use of a probiotic cocktail containing *Lactobacillus plantarum*, *Bifidobacterium animalis*, *Lactobacillus rhamnosus,* and *Lactobacillus acidophilus* with 7-week use, one capsule twice a day. In this study, the use of probiotics reduced OM severity and altered intestinal bacterial composition. Similarly, Jiang et al. [84] performed a randomised, double-blind, placebo-controlled trial in 93 patients with nasopharynx submitted to chemoradiotherapy. The intervention group that used the probiotics combination of *Bifidobacterium longum, Lactobacillus lactis*, and *Enterococcus faecium* had lower grades of OM. Similar results were found with the same study design in another cohort, Mirza et al. [85] reported a beneficial effect of the use of a *Bacillus clausii* suspension, prescribed twice a day for 30 days or until the end of RT, in 52 patients with HNC. Patients receiving the suspension had a later onset of OM and an average duration of ulceration shorter than the control group. Positive results were not observed in the other cohorts. De Sanctis et al. evaluated the use of *Lactobacillus brevis* CD2 in 75 patients undergoing HNC treatment. No difference in OM grade III/IV between the groups was reported or in other measurements of quality of life, such as body weight, general pain, and dysphagia [86]. As a result of this sustained research activity investigating the efficacy of probiotics for OM prevention and/or treatment, a recent systemic review and meta-analysis were able to confirm sufficient evidence to support their use during cancer therapy [87].

While these studies again reinforce the clinical utility of the microbiome in OM, it should be noted these studies were designed to influence OM via modulation of the gut microbiome. This highlights the difficulties in directly targeting the oral microbiome, a task that is complicated by the dynamic and hostile environment of the oral cavity that makes colonisation challenging but not impossible. For instance, strains of *Streptococcus dentisani*, an oral microbe that produces bacteriocins against oral pathogens and buffers acidic pH through an arginolytic pathway [88], were able to improve clinical and microbial parameters associated with oral health in healthy volunteers [89]. *Streptococcus salivarius* K12 is also a candidate oral probiotic. It is associated with oral health as it stimulates an anti-inflammatory response, modulates genes associated with adhesion to the epithelial layer and homeostasis and produces bacteriocin-like inhibitory substances [90]. Moreover, it is associated with lower OM in mice [91]. Therefore, investigating oral-directed probiotics, either alone or in conjunction with gut-directed probiotics, may offer a more substantial effect on OM prevention or control. Of interest is that oral microbiome transplantation (OMT)—the collection and application of a complex microbial ecosystem to the oral cavity—is emerging as a possible therapeutic strategy to support oral health. OMT is considered more advantageous compared with probiotics due to the diversity of species included (i.e., the whole ecosystem, not just selected strains). Although in its infancy, OMT is currently under investigation for the treatment of dental caries and periodontal disease [92]. In progressing with this treatment option, there are many practical considerations that must be defined or at least adequately justified, including the choice of the donor(s) and relevant eligibility considerations, microbial strain selection (if at all), sample collection, processing, dosing and administration.

As an alternative to direct microbial inputs, prebiotics are an alternative strategy to support commensal organisms-balancing and sustaining microbial homeostasis. Defined as a “food ingredient that beneficially affects the host by selectively stimulating the growth and/or activity of one or a limited number of bacteria, and thus improves host health” [93], prebiotics have also been largely applied with the goal of supporting the gut microbiome. Despite this, some prebiotics (e.g., nitrate) have been designed with the goal of improving oral health, helping to stabilise salivary pH, the oral biofilm and microbial eubiosis. Despite emerging evidence outlining the potential benefit of prebiotics for oral health, there have been no studies specifically investigating their ability to alter the development or progression of OM.

## 5. Conclusions

The oral and gut microbiome are diverse microbial ecosystems with the potential to influence various aspects of host physiology. During OM onset, changes in the oral microbiome are well identified and indeed arise in response to the profound and often hostile changes in the oral cavity during cancer treatment. These microbial changes then impact OM progression, dictating the depth of injury and duration of symptoms. As such, the hotly contested question of the “chicken or the egg” may, in fact, never be answered given the dynamic and bi-directional communication between the host and their resident microbes. Irrespective of cause and effect, it is clear that distinct microbial fingerprints influence an individual’s risk of OM and hold promise as predictive biomarkers. Similarly, augmenting the microbiome is highly feasible through direct and indirect strategies.

Numerous patient populations demonstrate the therapeutic potential of the oral and gut microbiome. These strategies may therefore deliver benefits that transcend a single symptom, offering substantive benefits to the overall quality of life of people living with or beyond cancer.

## Figures and Tables

**Figure 1 ijms-24-08274-f001:**
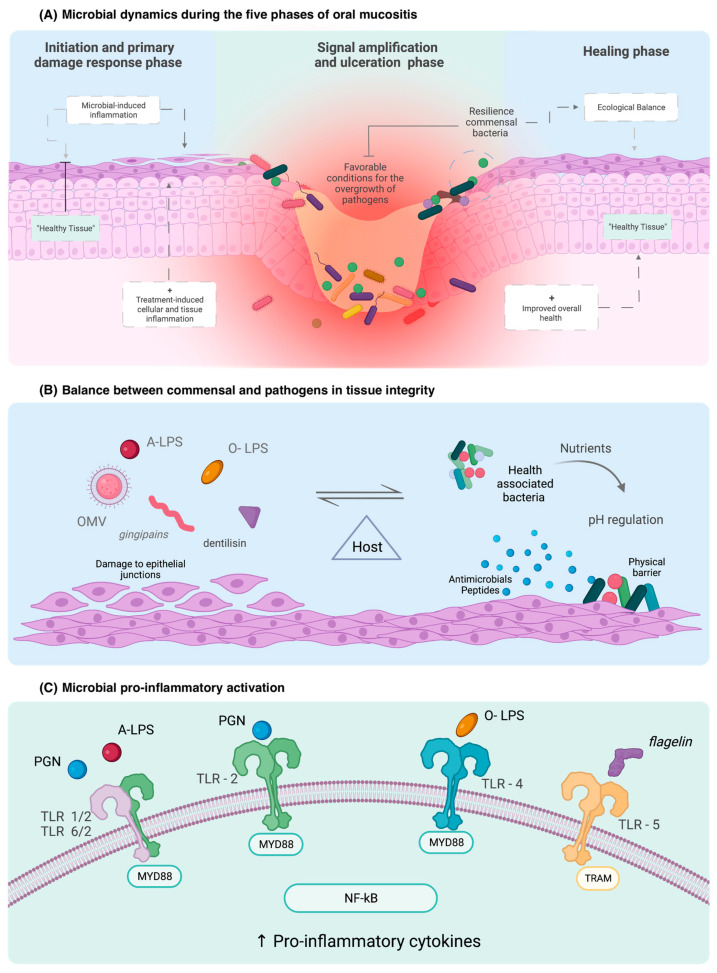
Role of the microbiome in epithelial cell integrity and immune activation in OM clinical course and development. OMV: outer membrane vesicle; PGN: peptidoglycan; O-LPS: O-antigen lipopolysaccharide; A-LPS: anionic lipopolysaccharide.

**Table 2 ijms-24-08274-t002:** Association of bacterial abundance and membrane TLR signalling.

Exogenous Ligands (Responsible for the Recognition)	Toll-like Receptor	Intracellular Adaptor Proteins	Signalling Pathways
Peptidoglycan (PGN) (G+) Glycolipid LAM (*Mycoplasma*) Peptideo PSMs (*Staphylococcus*)	2	myD88	Inflammatory cytokines
O-LPS (G-) Lipoteichoic acid (G+)	4	TIRAP → myD88 or TRAM	Inflammatory cytokines or Inflammatory cytokines IFN
Flagelin *(Treponema)*	5	myD88	Inflammatory cytokines
A-LPS (*Porphyromonas*) Peptidoglycan (PGN) (G+) Glycolipid LAM (*Mycoplasma*) Peptideo PSMs (*Staphylococcus*)	1/2 *	myD88	Inflammatory cytokines
	2/6 *	myD88	Inflammatory cytokines

* heterodimer TLR; PGN: peptidoglycan; G+: Gram positive bacteria; G-: Gram negative bacteria; LAM: Lipoarabinomannan; PSMs: Phenol-soluble modulins; myD88: myeloid differentiation primary-response protein 88; O-LPS: O-antigen polysaccharide; TIRAP: TIR domain-containing adaptor protein; TRAM: Trif-related adapter molecule; A-LPS: anionic polysaccharide.

## Data Availability

Data sharing not applicable.

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
