# Peer review of "From Pathogenesis to Intervention: The Importance of the Microbiome in Oral Mucositis"

_ijms, 2023, doi:10.3390/ijms24098274_

Round 1
Reviewer 1 Report
Manuscript of considerable interest for the dental sector, before a correct evaluation of a possible publication it needs a major revision
Abstract: to better highlight the data collected
Keywords few add specific ones that are registered on MeSH
Introduction: insert and all systems based on probiotics, postbiotics and paraprobiotics to reduce the incidence of bacterial load on the periodontium and mucous membranes already studied by Prof Scribante's research group.
Materials and methods How was the sample size calculated?
Results, very confusing reorganize the tables to allow the reader to understand them better
Discussion: add as future objectives the use of ozone and ozonated agents and photodynamic therapy to reduce the bacterial load still studied by the research group of Prof Scribante
Conclusions; add proactive action
Bibliography; add references required
Author Response
Dear reviewers,
Thank you for giving us the opportunity to submit a revised draft of our manuscript. We appreciate the time and effort that you have dedicated to providing your valuable feedback. We have been able to incorporate changes to reflect the suggestions provided by the reviewers.
Here is a point-by-point response to the reviewers’ comments and concerns.
Comments from Reviewer 1
- Abstract: to better highlight the collected data
We appreciate the suggestion. The data collection strategy was added in the abstract (lines 24-27, page 1).
- Keywords few add specific ones that are registered on MeSH
The former keywords (microbiome; microbiota; oral mucositis; biomarker; supportive care; radiation; chemotherapy) were replaced, according to Mesh registration, for the new keywords: microbiota; stomatitis; biomarkers; radiotherapy; chemotherapy (line 31, page 1).
- Introduction: insert and all systems based on probiotics, postbiotics and paraprobiotics to reduce the incidence of bacterial load on the periodontium and mucous membranes already studied by Prof Scribante's research group.
We thank the Reviewer for pointing out Prof. Scribante's excellent line of research; the indicated references have been added in section 4 of the text, alongside discussions of viable therapies in non-cancer patients (lines 473-478, page 12).
- Materials and methods How were the sample size calculated?
This is a review compiling articles in English available in the PubMed (MEDLINE) database in the oral and gut microbiome (with 16S rRNA sequencing methodology) analysing the bacterial shift during the clinical course of oral mucositis. To maintain a broad understanding of the topic, recent works with sequencing methodology that correlated the human microbiome with the clinical course of oral mucositis were included. We are not sure what exactly you are referring to with respect to the sample size calculation given this is a narrative review.
- Results, very confusing reorganize the tables to allow the reader to understand them better.
We appreciate the suggestion. We have organized the headers of Table 1 and Table 2. The main results of Table 1 were rewritten for a better understanding of the key findings in each article (Pages 4, 5 and 8).
- Discussion: add as future objectives the use of ozone and ozonated agents and photodynamic therapy to reduce the bacterial load still studied by the research group of Prof Scribante
In conjunction with the viable therapies described for non-oncology patients, the use of ozone and photobiomodulation were described in the text (lines 473-478, page 12).
- Conclusions; add proactive action.
After proofreading, the passive voice is not used in the conclusion section. Greater emphasis on the clinical applicability of using the microbiome as a predictive tool for oral mucositis has been added (lines 533-547, page 14).
- Bibliography; add references required.
The references were updated according to the new references and proofreading.
Reviewer 2 Report
I am honored to review a manuscript titled “From pathogenesis to intervention: the importance of the microbiome in oral mucositis” proposed by Bruno et al.
The manuscript was well-written and very informative. However, I suggest a thorough proofreading of the WHOLE text. There are plenty of language errors, a sample of which is provided below.
Introduction
haematopoietic stem cell transplant should be haematopoietic stem cell transplantation
on patient quality of life should be on the patient’s quality of life
The oral microbiome and its influence on oral health
In terms of the microbial composition should be In terms of microbial composition
3.2. Evidence of causal involvement of the oral microbiome to oral mucositis (replace to oral… by in oral…)
3.3. Mechanisms by which the microbiome contribute to oral mucositis (replace contribute … by contributes …
Author Response
Dear reviewers,
Thank you for giving us the opportunity to submit a revised draft of our manuscript. We appreciate the time and effort that you have dedicated to providing your valuable feedback. We have been able to incorporate changes to reflect the suggestions provided by the reviewers.
Here is a point-by-point response to the reviewers’ comments and concerns.
Comments from Reviewer 2
- I am honored to review a manuscript titled “From pathogenesis to intervention: the importance of the microbiome in oral mucositis” proposed by Bruno et al.
The manuscript was well-written and very informative. However, I suggest a thorough proofreading of the WHOLE text. There are plenty of language errors, a sample of which is provided below.
Introduction
haematopoietic stem cell transplantation should be haematopoietic stem cell transplantation
on patient quality of life should be on the patient's quality of life
- The oral microbiome and its influence on oral health
In terms of the microbial composition should be
3.2. Evidence of causal involvement of the oral microbiome to oral mucositis (replace to oral… by in oral…)
3.3. Mechanisms by which the microbiome contribute to oral mucositis (replace contribute … by contributes …
We thank the Reviewer for the thorough and positive review. We are grateful for the Reviewer's comment. The whole text was proofread and updated.